# High prevalence of heteroresistance in *Staphylococcus aureus* is caused by a multitude of mutations in core genes

**Sheida Heidarian, Andrei Guliaev, Hervé Nicoloff, Karin Hjort***, Dan I. Andersson*

Department of Medical Biochemistry and Microbiology, Uppsala University, Uppsala, Sweden

* karin.hjort@imbim.uu.se (KH); dan.andersson@imbim.uu.se (DA)

**Data Availability Statement:** All relevant data are within the paper and its Supporting Information files.

## Abstract

Heteroresistance (HR) is an enigmatic phenotype where, in a main population of susceptible cells, small subpopulations of resistant cells exist. This is a cause for concern, as this small subpopulation is difficult to detect by standard antibiotic susceptibility tests, and upon antibiotic exposure the resistant subpopulation may increase in frequency and potentially lead to treatment complications or failure. Here, we determined the prevalence and mechanisms of HR for 40 clinical *Staphylococcus aureus* isolates, against 6 clinically important antibiotics: daptomycin, gentamicin, linezolid, oxacillin, teicoplanin, and vancomycin. High frequencies of HR were observed for gentamicin (69.2%), oxacillin (27%), daptomycin (25.6%), and teicoplanin (15.4%) while none of the isolates showed HR toward linezolid or vancomycin. Point mutations in various chromosomal core genes, including those involved in membrane and peptidoglycan/teichoic acid biosynthesis and transport, tRNA charging, menaquinone and chorismite biosynthesis and cyclic-di-AMP biosynthesis, were the mechanisms responsible for generating the resistant subpopulations. This finding is in contrast to gram-negative bacteria, where increased copy number of bona fide resistance genes via tandem gene amplification is the most prevalent mechanism. This difference can be explained by the observation that *S. aureus* has a low content of resistance genes and absence of the repeat sequences that allow tandem gene amplification of these genes as compared to gram-negative species.

## Introduction

During the last decades, research on mechanisms of antibiotic resistance has focused on stable genetic mutations or acquisition of antibiotic resistance genes, which typically confer phenotypic resistance to all cells within a population [1,2]. Thus, the presence of resistance genes or mutations conferring resistance is resulting in a resistance phenotype and there is a strong correlation between genotype and observed phenotype [3]. However, some exceptions exist that complicate the detection of resistant bacteria, and one such example is heteroresistance (HR), where a small subpopulation of resistant bacteria exist in a major population of susceptible bacteria [4]. Different from antibiotic tolerance or persistence, HR subpopulations can

**Funding:** This project was financed by funds granted to DIA by NIH (grant 1U19AI158080-01), Swedish research Council (grant 2021-02091) and Wallenberg Foundation (grant 2018.0168). The funders had no role in study design, data collection and analysis, decision to publish, or preparation of the manuscript

**Competing interests:** The authors have declared that no competing interests exist

**Abbreviations:** AST, antimicrobial susceptibility test; BHI, brain heart infusion; CBP, clinical breakpoint; EUCAST, European Committee on Antimicrobial Susceptibility Testing; FnBP, fibronectin binding protein; HR, heteroresistance; hVISA, heterogenous vancomycin-intermediate *Staphylococcus aureus*; LPG, lysyl-phosphatidylglycerol; MET, macro E-test; MHA, Mueller–Hinton agar; MHB, Mueller–Hinton broth; MIC, minimum inhibitory concentration; PAP, population analysis profiling.

proliferate under antibiotic selection [2,5,6]. The low frequency makes it difficult for routine antimicrobial susceptibility tests (ASTs) [4,7] to reliably detect the resistant subpopulation and correctly determine the susceptibility profile of a bacterial isolate.

Although more than 70 years have passed since the first description of heterogeneous resistance in gram-negative isolate [8] and over 50 years for gram-positive bacteria [9], there is still a lack of studies regarding the prevalence of HR in many bacterial pathogens, the genetic mechanisms behind HR, and the clinical impact of HR on antibiotic treatment outcomes. HR has been reported for several bacteria and antibiotic combinations [2,4]; however, the lack of a clear and precise definition and standard method to identify HR complicates the interpretation and comparison of published studies on HR prevalence [4]. Despite the fact that there are different approaches and terminologies to determine HR, the population analysis profiling (PAP) test is considered the current "gold standard" for phenotyping HR [7]. Using the PAP test, HR was defined here by the presence of subpopulations ($\geq 10^{-7}$) of bacteria whose resistance was $\geq$8-fold above the minimum inhibitory concentration (MIC) [4,10]. The rationale behind using the frequency cutoff of $1 \times 10^{-7}$ is to distinguish HR from typical resistance frequencies, which are usually lower [11]. Regarding effects of HR on treatment outcome, studies in gram-negative and gram-positive bacteria suggest that the presence of resistant subpopulations can lead to increased mortality in animal studies [12–15] and to either prolonged hospitalization or an increased mortality rates [16–19] (see also below regarding heterogenous vancomycin-intermediate *Staphylococcus aureus* (hVISA)).

A few different mechanisms underlying the HR phenotype have been described and they can be divided into stable and unstable HR. Resistance selected from stable HR isolates is maintained following removal of antibiotic selection pressure, while resistance emerging from unstable HR isolates reverts in absence of selection pressure [2]. In gram-negative bacteria, 2 main genetic mechanisms have been observed that can cause unstable HR [10,20]. The most common mechanism generating resistant subpopulations is tandem gene amplifications, where increased dosage of various bona fide resistance genes present in the genome causes increased resistance. Tandem gene amplifications in bacteria typically arise from spontaneous RecA-dependent homologous recombination between repeated sequences (e.g., rRNA operons, IS elements, transposases) in sister chromatids at replication, and further unequal cross-over events within the duplication can either lead to further amplification or its loss. For this mechanism, the instability of the resistance phenotype results from the intrinsic genetic instability of tandem amplifications and the cost of carrying and expressing the amplified genes [10,20]. The second mechanism involves genetically stable resistance mutations that confer a high fitness cost, and in the absence of an antibiotic the high fitness cost leads to a selection of second-site compensatory mutations that reduce the cost, which sometimes is associated with reversion to susceptibility [10].

The majority of HR studies focus on gram-negative pathogens [10,18,21–24] and less data are available about HR prevalence and mechanisms in gram-positive bacteria. To address this lack of gram-positive studies, we examined HR in *S. aureus*, a common bacterial pathogens causing a variety of infections [25]. The majority of HR studies in *S. aureus* have focused on the prevalence of hVISA [26–28] but some recent reports have studied occurrence of HR for daptomycin [29], ceftaroline [30], teicoplanin [31], trimethoprim-sulfamethoxazole [32], and oxacillin [33]. Since many factors varied between these studies (e.g., type of antibiotic studied, geographical and infection-type origin of the bacterial isolates and differences in the methodologies used to determine HR), it is not surprising that the prevalence of HR in *S. aureus* can vary extensively across studies. For example, hVISA prevalence frequencies in isolates from different countries using the same methodology varied from 0% to 13.7% [26,34–36]. Additionally, the prevalence of HR can also vary strongly depending on the examined class of

antibiotic [30,32]. The impact of HR on treatment outcome has been quite extensively studied for hVISA with some case reports and retrospective studies [14,37–39] reporting that hVISA is correlated with worse outcomes, but other reports suggest that the HR phenotype does not affect treatment outcome [36,40–42].

In this work, we measured the prevalence and identified potential HR mechanisms against different clinically relevant classes of antibiotics for 40 clinical *S. aureus* isolates utilizing a consistent methodology and HR definition when evaluating the isolates against different clinically relevant classes of antibiotics. We demonstrate that: (i) the prevalence of HR differs widely among different antibiotic classes; (ii) the frequency of HR where the resistant isolates reach MICs above the clinical breakpoint (CBP) is substantial for some antibiotics; (iii) point mutation in various chromosomal core genes is the main molecular mechanism generating the resistance in our set of HR isolates; and (iv) the low frequency of resistance genes and lack of flanking tandem repeat sequences explain why gene amplification-mediated HR is not observed in *S. aureus*.

## Material and methods

### Bacterial strains, antibiotics, and growth conditions

Forty clinical isolates of *S. aureus* originating from wound and blood samples were randomly collected from clinical microbiology laboratories in Denmark (Rigshospitalet) Norway (University Hospital of North Norway), Spain (Hospital Ramon y Cajal), and Sweden (Karolinska University Hospital) (Table 1) in 2020. The 6 different antibiotics used in this study, daptomycin (DAP), gentamicin (GEN), linezolid (LNZ), oxacillin (OXA), teicoplanin (TEC), and vancomycin (VAN), were purchased from Sigma-Aldrich. For MIC determination, prescreening, PAP test, growth rate, and mutation rate analysis Mueller–Hinton broth (MHB) and Mueller–Hinton agar (MHA) (Difco, Becton Dickinson (BD)) were used as a growth medium, except for VAN studies, where brain heart infusion (BHI) broth and agar (Difco, Becton Dickinson) was used for prescreening and PAP analysis. All incubations were carried out at 37˚C with shaking at 190 rpm for broth cultures.

### Determination of the minimum inhibitory concentration

MICs for each antibiotic and isolate was determined by E-test according to manufacturer's recommendations. Briefly, 0.5 McFarland suspension of each bacterial isolate was used as inoculum and spread using sterile cotton swab on MHA plates, followed by the application of the E-test (bioMérieux). After 24 h of incubation of the plates at 37˚C, the MIC value for each clinical isolate was determined according to European Committee on Antimicrobial Susceptibility Testing (EUCAST) guidelines. *S. aureus* ATCC 29213 was used as a control strain for the E-test determination. MIC values and distributions are shown in Tables 1 and S1 and S1 Fig.

### Prescreening of susceptible clinical isolates

All susceptible isolates were prescreened for growth of resistant subpopulations for each isolate/antibiotic combination. For DAP and OXA, the prescreen was conducted using MHA plates enriched with 50 mg/L of $Ca^{2+}$ (Sigma-Aldrich) and 2% NaCl (VWR-Chemicals), respectively. One single colony was inoculated in 100 μl of MHB in a 96-well microtiter plate. After the first overnight growth, 1 μl of the bacterial suspension from a $10^{-2}$ dilution in saline solution was transferred to 1 ml of MHB medium in a 10 ml sterile plastic tube and incubated for an additional 24 h. Approximately $10^8$ CFUs from the overnight cultures were plated on MHA plates containing antibiotics at 4, 8, and 16-fold above the MICs of each antibiotic and

**Table 1. Minimal inhibitory concentrations (mg/L) of 6 clinically relevant antibiotics for 40 clinical *S. aureus* isolates.** EUCAST CBPs for each antibiotic utilized for MIC determination is: (DAP (daptomycin): >1 mg/L), (GEN (gentamicin): >2 mg/L), (LNZ (linezolid): >4 mg/L), (OXA (oxacillin): >2 mg/L), (TEC (teicoplanin): >2 mg/L), and (VAN (vancomycin): >4 mg/L). The MIC value for each resistant isolate to specific antibiotics is indicated in bold.

| Isolates DA number | Country of origin | DAP | GEN | LNZ | OXA | TEC | VAN |
|---|---|---|---|---|---|---|---|
| DA70300 | Spain | 0.38 | 0.75 | 0.75 | 0.25 | 0.38 | 0.75 |
| DA70302 | Spain | 0.38 | 0.25 | 2 | 0.38 | 0.75 | 1 |
| DA70314 | Spain | 0.38 | 0.19 | 0.75 | 0.5 | 0.38 | 1 |
| DA70318 | Spain | 0.5 | **32** | 4 | 0.5 | 1 | 1.25 |
| DA70322 | Spain | 0.5 | 0.25 | 3 | 0.25 | 0.25 | 1.5 |
| DA 70324 | Spain | 0.125 | 0.25 | 1 | 0.5 | 0.38 | 0.75 |
| DA 70338 | Spain | 0.19 | 0.38 | 2 | **48** | 0.75 | 1.5 |
| DA 70348 | Spain | 0.5 | 0.5 | 1 | 0.38 | 0.5 | 1 |
| DA 70350 | Spain | 0.5 | 0.25 | 2 | **64** | 1.5 | 1.5 |
| DA 70352 | Spain | 0.25 | 0.19 | 0.75 | 0.25 | 0.38 | 1 |
| DA 70484 | Sweden | 0.125 | 0.25 | 1 | 0.25 | 0.25 | 1 |
| DA 70488 | Sweden | 0.75 | 0.125 | 1 | 0.25 | 0.5 | 1 |
| DA 70500 | Sweden | 0.38 | 0.19 | 0.5 | 0.25 | 0.5 | 0.75 |
| DA 70504 | Sweden | 0.19 | 0.25 | 2 | 0.5 | 0.5 | 1 |
| DA 70506 | Sweden | **4** | 0.75 | 2 | 0.5 | **3** | 1.5 |
| DA 70512 | Sweden | 0.25 | 0.38 | 2 | 0.38 | 0.38 | 0.5 |
| DA 70516 | Sweden | 0.5 | 0.38 | 1 | 0.5 | 0.75 | 1.25 |
| DA 70518 | Sweden | 0.19 | 0.25 | 0.75 | 0.38 | 0.38 | 1 |
| DA 70520 | Sweden | 0.5 | 0.38 | 3 | 0.5 | 0.5 | 1 |
| DA 70524 | Sweden | 0.5 | 0.38 | 0.75 | 0.19 | 1 | 1 |
| DA 70672 | Norway | 0.5 | 0.38 | 3 | 0.38 | 0.75 | 1.25 |
| DA 70674 | Norway | 0.25 | 0.3 | 0.75 | 0.25 | 0.25 | 0.75 |
| DA 70682 | Norway | 0.38 | 0.19 | 3 | 1 | 0.75 | 1 |
| DA 70684 | Norway | 0.25 | 0.25 | 0.75 | 0.5 | 0.75 | 1 |
| DA 70686 | Norway | 0.5 | 0.25 | 1 | 0.25 | 1.5 | 1.25 |
| DA 70692 | Norway | 0.19 | 0.125 | 1.5 | 0.38 | 0.38 | 0.75 |
| DA 70694 | Norway | 0.38 | 0.25 | 1 | 0.75 | 1 | 1 |
| DA 70700 | Norway | 0.19 | 0.125 | 1 | 0.25 | 0.5 | 1 |
| DA 70708 | Norway | 0.25 | 0.3 | 2 | 0.38 | 0.75 | 1 |
| DA 70710 | Norway | 0.094 | 0.125 | 0.75 | 0.125 | 0.25 | 0.38 |
| DA 70866 | Denmark | 0.125 | 0.25 | 0.75 | 0.19 | 0.75 | 0.75 |
| DA 70870 | Denmark | 0.5 | 0.25 | 0,75 | **64** | 1 | 1 |
| DA 70876 | Denmark | 0.5 | 0.38 | 2 | 0.5 | 0.38 | 1 |
| DA 70880 | Denmark | 0.064 | 0.094 | 0.5 | 0.25 | 0.38 | 0.38 |
| DA 70890 | Denmark | 0.25 | 0.3 | 1 | 0.75 | 0.5 | 1 |
| DA 70896 | Denmark | 0.19 | 0.19 | 1 | 0.75 | 0.25 | 0.75 |
| DA 70898 | Denmark | 0.38 | 0.25 | 3 | 0.75 | 0.38 | 1 |
| DA 70900 | Denmark | 0.5 | 0.19 | 1.5 | 1 | 2 | 1 |
| DA 70906 | Denmark | 0.5 | 0.064 | 3 | 0.5 | 0.5 | 1.5 |
| DA 70912 | Denmark | 0.25 | 0.19 | 1.5 | 0.19 | 0.75 | 1 |

CBP, clinical breakpoint; EUCAST, European Committee on Antimicrobial Susceptibility Testing; MIC, minimum inhibitory concentration.

individual isolate. To determine the total cell counts, 5 μl drops of $10^{-4}$ to $10^{-7}$ dilutions of the overnight culture in saline was placed in duplicates on agar plates without antibiotics and incubated for 72 h. For all isolates/antibiotic combinations, the frequency of bacteria growing on

each antibiotic plate was calculated by dividing CFUs on plates with antibiotics with CFUs of the total number of viable bacteria on media without antibiotics (S2 Fig). All tested isolates with a subpopulation frequency above $10^{-7}$ growing at 8-fold above MIC were categorized as possible HR isolates and PAP tested for confirmation of HR phenotype. Two biological replicates were used for each isolate.

## Population analysis profile (PAP test)

PAP tests were performed for all parental isolates that were selected as potential HR isolates in the prescreen as follows: After 24 h of incubation of a single bacterial colony in 100 μl of MHB medium in a 96-well microtiter plate, cultures were diluted in saline solution to $10^{-3}$ and 1 μl from this dilution was transferred (approximately 1,000 cells) to 1 ml MHB in 10 ml sterile plastic vial and further incubated. This procedure assures that no preexisting mutants are present in the inoculum and that the mutants emerge during the last incubation step (corresponding to approximately 17 generations of growth, i.e., from $10^3$ to $10^8$ cells). After the 24 h incubation, $10^8$ bacterial cells from the overnight culture were placed as 5 μl drops (plates supplemented with antibiotics at concentrations 0.25, 0.5, and 1 × MIC) or spread with sterile glass beads (plates supplemented with antibiotics at concentrations 2, 4, 8, and 16 × MIC) on freshly prepared MHA plates (individual MIC values for each isolate obtained using E-test). For the control plate, overnight cultures were serially diluted to $10^{-7}$ and 5 μl drops from dilution $10^{-4}$ to $10^{-7}$ were placed in duplicates on plates without antibiotics. Following 72 h of incubation at 37°C, colonies were counted. The frequency of the subpopulation for each antibiotic concentration was calculated using the following formula: CFUs of bacteria on antibiotic-containing plates/CFUs of bacteria on plats without antibiotics. The frequency of bacteria was plotted against each antibiotic concentration using GraphPad Prism (GraphPad; San Diego, California, United States of America). Experiments were performed in triplicates for all isolates.

## Isolation of resistant clones growing above MIC

For each antibiotic where HR was detected, resistant mutants were selected from 5 different isolates displaying HR to the antibiotic. Each selection was performed using 5 biological replicates. Briefly, $10^8$ bacterial cells were plated on MHA plates containing antibiotic concentrations 8-fold above MIC. Mutants growing as colonies were picked at both 48 h and 72 h of incubation. All colonies were re-streaked on MHA plates with the same antibiotic concentration, followed by inoculation of each clone into 2 ml MHB (in a 10 ml sterile plastic tube) supplemented with the same concentration of antibiotics. After overnight incubation, the 1 ml of bacterial culture was frozen at −80°C in 10% DMSO for the strain collection, and 1 ml was centrifuged and the bacterial cell pellet was preserved at −20°C for subsequent DNA extraction and whole-genome sequencing analysis.

## Whole-genome sequencing and identification of mutations

Whole-genome sequencing was done on the parental HR clinical isolates and the isolated resistant clones. DNA was extracted from 1 ml of overnight cultures (resistant clones were grown under antibiotic selection) using the MasterPure Gram Positive DNA Extraction Kit (Lucigen), following the manufacturer's instructions with small modification in the cell digestion process by adding 2 μl of Lysostaphin enzyme from *Staphylococcus simulans* (Sigma-Aldrich) to the cell lysis solution. The parental HR clinical isolates were sequenced with both Nanopore (Oxford Nanopore Technologies, United Kingdom) in-house (using the rapid barcoding kit 96 in order to multiplex 40 genomes per each R9 cells; sequencing performed on

MinION Mk1C), and with DNBseq (800 base pairs paired-end libraries, 50× coverage on average) at BGI (Warsaw, Poland). The isolated resistant clones were sequenced with DNBseq only. The quality of sequencing reads was assessed using FastQC v0.11.91 and MultiQC v1.122. Short reads were trimmed using fastp v0.20.13. Nanopore reads shorter than 1,000 bp were removed using Filtlong v.0.2.14. The filtered reads were assembled using Unicycler v0.4.85 if Nanopore coverage was below 20×; otherwise, a custom pipeline involving Flye v2.9.16, medaka v1.6.07, Polypolish v0.5.08, BWA v0.7.17 [43], and seqkit v2.2.0 was used. The bioinformatics analysis was implemented using Snakemake workflow management system v7.24.0. and is available on GitHub (github.com/andrewgull/saureus_hr). To identify protein-coding genes and tRNA genes, the assemblies were automatically annotated through Prokka v1.14.69 and Prodigal v2.6.310, along with tRNA scan-SE v2.0.911. The DNBseq reads from the selected resistant clones were mapped onto the reference genome of the corresponding parental HR clinical isolate they were derived from, using CLC Genomic Workbench software (Qiagen). Mutations and amplifications were detected using basic variant detection, InDels, structural variants, and coverage analysis with the CLC software. Potential antibiotic resistance genes were identified using the CARD online database, with the preset selection criteria (strict cutoff hits with an identity percentage of reference sequence >90% and a percentage length of reference sequence >90%). Repeat analysis was performed through manual inspection to check for the presence of repeats within a distance of up to 200 kbp from the target resistance gene, using the CLC program. To analyze protein–protein interactions network and functional enrichment, the STRING database was used [44]. Sequence data are deposited at NCBI as PRJNA1036259.

## Fitness-cost and resistance analysis of mutants

The maximum exponential growth rates were measured using a Bioscreen analyzer (Oy Growth Curves Ab). Five independent cultures for each mutant and parental strain were grown overnight and diluted to $5 \times 10^6$ cfu/ml in 1 ml fresh MHB medium the next day, then 300 μl of each cell suspension was transferred in duplicate to each well of a honeycomb plate to begin the experiment. The plates were incubated in the Bioscreen C analyzer at 37˚C with continues shaking at a medium amplitude for 24 h. Readings were performed at an optical density of 600 nm with a 5-min measurement interval and 5 s pause in the shaking just prior to each measurement. The maximum exponential growth rate was calculated from absorbance values between 0.02 and 0.09, with an R-value above 0.999, using BAT 2.0 [45]. The relative growth rate for each isolated clone was normalized to their respective parental HR isolate, which was set to 1. Five biological and 2 technical replicates were used for all isolated clones and parental HR isolates. The MIC of each isolated clone was determined by a single E-test after 24- or 48-h incubation depending on the growth rate of the clone.

## Mutation rate analysis

The mutation rate was determined for all HR isolates using Luria–Delbrück fluctuation test and calculated according to the MSS Maximum Likelihood method using the FALCOR (Fluctuation AnaLysis CalculatOR) online tool [46]. First, 5 overnight cultures were started by inoculating 5 independent colonies in $5 \times 100$ μl of MHB medium in a 96-well microtiter plate. Following overnight growth at 37˚C under vigorous shaking, the culture was diluted 100-fold in saline, and 1 μl from each diluted culture (corresponding to about $10^3$ bacterial cells) was inoculated in $4 \times 1$ ml of MHB medium, resulting in 20 replicates per isolate. After 24 h of incubation at 37˚C under vigorous shaking, 10 μl to 800 μl of culture (depending on the isolates) was plated on MHA plates with 100 mg/L of rifampicin [47]. Plates were incubated for

48 h in the dark at 37°C. To determine total number of cells, each overnight culture was serially diluted in saline and 5 μl drops from $10^{-4}$ to $10^{-7}$ dilutions were plated in duplicates on non-antibiotic-containing plates. After 48 h of incubation, colonies on plates with and without antibiotics were counted.

### Statistical analysis

Spearman's correlation analysis was used to assess the association between the relative growth rate and the MIC value of the resistant mutants. The correlation between the MIC value and the presence of the HR phenotype was examined using the Mann–Whitney U test. To compare the mutation rates between different HR and non-HR groups, we conducted a one-way ANOVA test. All statistical analyses were performed using GraphPad Prism (9.0.1). Two-sided tests were employed, and a *P*-value of less than 0.05 was considered statistically significant.

## Results

### MICs of 6 clinically relevant antibiotics

MICs of daptomycin (DAP), gentamicin (GEN), linezolid (LNZ), oxacillin (OXA), teicoplanin (TEC), and vancomycin (VAN) were determined for all 40 parental isolates (Table 1). For HR determination, only susceptible isolates (based on the EUCAST clinical breakpoint guideline, 2023) were used and 6 isolate/antibiotic combinations (DA70506 DAP, TEC), (DA70318 GEN), (DA70338, DA70350 and DA70870, OXA) were therefore excluded from further investigation. The distribution of MIC values varied between antibiotics, and some antibiotics such as LNZ and TEC showed a wide distribution of MIC values (S1 Fig and S1 Table). The MIC distributions for DAP, GEN, OXA, TEC, and VAN resembled the datasets provided in the EUCAST guideline (collected from more than 30,000 samples worldwide) [48], while for LNZ the MIC distributions were slightly lower.

### Prevalence of HR varies depending on antibiotic

To determine the prevalence (i.e., the % fraction of parental isolates that showed the HR phenotype) of HR among the 40 original parental isolates, we performed prescreens with all isolate/antibiotic combinations where the isolate was susceptible towards DAP, GEN, OXA, LNZ, TEC, or VAN (234 isolate and antibiotic combinations, S2 Fig) by plating $10^8$ cells on agar plates containing the respective antibiotics at 4, 8, and 16× MIC. Each clinical isolate that had a resistant subpopulation at a frequency of $\geq 1 \times 10^{-7}$ growing at 8-fold above MIC was considered potentially HR and was further analyzed by a full PAP test (Fig 1, S3 Fig and S1 and S2 Tables). The concordance between prescreen and PAP test was high for some antibiotics but low for others. Thus, for DAP and GEN there was perfect concordance between the prescreen and the PAP test with 25.6% and 69.2% of the isolates, respectively, showing HR (Table 2). For OXA and TEC, there was reasonable concordance with frequencies of 29.7% and 20.5%, respectively, in the prescreen but with slightly lower frequencies determined from PAP tests, 27% and 15.4%, respectively (Table 2 and S3 Fig). However, for VAN and LNZ no concordance was seen since all the potential HR isolates identified in the prescreen could not be confirmed in the PAP test and they were therefore considered false positives. We did not observe any correlation between geographic site of collection and presence of HR phenotype (however, it should be noted that the number of isolates was relatively low and a potential correlation could easily have been missed).

Since it is likely that an HR isolate with subpopulations of resistant cells that grow beyond the CBP are more problematic to treat, we also determined these frequencies. As can be seen

**Table 2. Prevalence (% fraction of parental isolates) of heteroresistance among the susceptible clinical *S. aureus* parental isolates for 6 different antibiotics.** DAP (daptomycin), GEN (gentamicin), LNZ (linezolid), OXA (oxacillin), TEC (teicoplanin), and VAN (vancomycin).

| Antibiotic | Number of susceptible parental isolates analyzed | Number (percentage) of parental isolates showing HR in prescreen | Number (percentage) of parental isolates showing HR by PAP test | Number (percentage) of parental isolates with subpopulations resistant above EUCAST CBP |
|---|---|---|---|---|
| DAP | 39 | 10 (25.6%) | 10 (25.6%) | 3 (7.7%) |
| GEN | 39 | 27 (69.2%) | 27 (69.2%) | 11 (28.2%) |
| LNZ | 40 | 4 (10%) | 0 (0%) | 0 (0%) |
| OXA | 37 | 11 (29.7%) | 10 (27%) | 1 (2.7%) |
| TEC | 39 | 8 (20.5%) | 6 (15.4%) | 2 (5.1%) |
| VAN | 40 | 14 (35%) | 0 (0%) | 0 (0%) |

CBP, clinical breakpoint; EUCAST, European Committee on Antimicrobial Susceptibility Testing; HR, heteroresistance; PAP, population analysis profiling.

in Table 2, among all susceptible clinical *S. aureus* isolates, 7.7% (3/39), 28.2% (11/39), 2.7% (1/37), and 5.1% (2/39) of HR strains for DAP, GEN, OXA, and TEC respectively, had resistant subpopulations reaching above the CBPs. Furthermore, there was a relatively high frequency of isolates that showed HR to several drugs. Thus, among the 40 *S. aureus* isolates, 15% (6/40 isolates) were non-HR to all 6 antibiotics, whereas 52.5% of the isolates (21/40) exhibited HR to 1 antibiotic, 17.5% (7/40) towards 2 antibiotics, and 15% (6/40) were HR towards 3 drugs (Figs 1 and S4 and S2 and S3 Tables).

There were no statistically significant differences between HR and non-HR isolates in regards to their MIC towards OXA and GEN, while for DAP and TEC the HR isolates showed a significantly lower MIC than the non-HR isolates. However, the difference was not pronounced and there was overlap between the 2 groups (S5 Fig).

## Resistance profile and fitness cost of antibiotic-resistant mutants selected from the HR subpopulation

Mutants that grew at ≥8-fold above the MICs were selected from 5 parental HR isolates for each of the following antibiotics: DAP (13 mutants), GEN (17 mutants), OXA (17 mutants), and TEC (21 mutants). The resistance profiles and fitness costs were then determined for all mutants. In comparison to their respective parental isolate, the majority of mutants displayed MIC values that were at least 4 times higher than those of their cognate parental isolates. Almost without exceptions, the resistant mutants showed a reduction in fitness with average fitness costs corresponding to 37%, 40%, 30%, and 22% reductions in exponential growth rates for mutants resistant to DAP, GEN, OXA, and TEC, respectively (S6 Fig and S1 Table). There was a correlation between fitness cost and MIC values for all mutants selected for GEN (*p*-value 0.004), but not for DAP, OXA, and TEC (*p*-values 0.086, 0.088, and 0.056, respectively) (S7 Fig and S1 Table).

## Whole-genome sequencing analysis

To try to understand the mechanism causing resistance in the subpopulations, i.e., how is the HR phenotype generated, whole-genome sequencing was performed on 68 resistant mutants selected on DAP, GEN, OXA, and TEC and the corresponding 10 different parental isolates. The number of mutations per mutant varied between 1 to 8 (S8–S11 Figs). For GEN, most of the mutants selected had 1 mutation per isolate, while mutants selected on DAP, OXA, and TEC had on average 2, 4, and 2 mutations, respectively. Figs 2 and 3 provide schematic representations and frequencies of genes mutated in the different resistant mutants for each antibiotic and their respective functions in various cellular pathways.

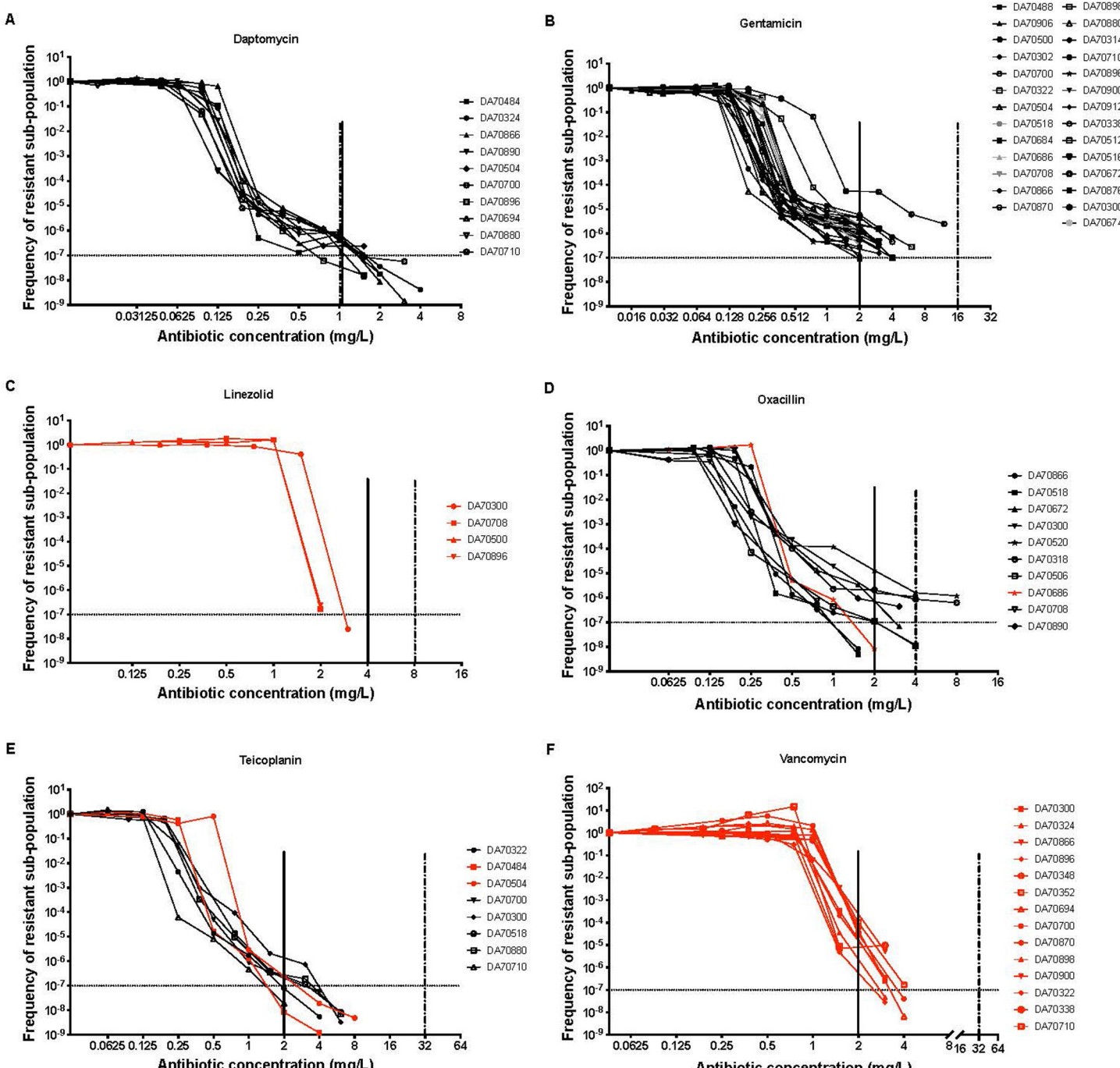

**Fig 1. Population analysis profiles (PAP).** Compiled PAP curves of all potential HR isolates identified in the prescreen (growth of a subpopulation $\geq 1 \times 10^{-7}$ on 8× above the MIC) for 6 antibiotics. The red lines in the graph show non-HR isolates. Each curve is an average of 3 independent experiments. The solid vertical lines indicate the CBP according to EUCAST. The half-dotted vertical lines indicate the CBP according to CLSI guidelines and the horizontal dotted lines indicate the subpopulation frequency cutoff ($1 \times 10^{-7}$) used to identify HR isolates. (A) DAP (daptomycin), (B) GEN (gentamicin), (C) LNZ (linezolid), (D) OXA (oxacillin), (E) TEC (teicoplanin), and (F) VAN (vancomycin). For raw data, see S2 Table. CBP, clinical breakpoint; EUCAST, European Committee on Antimicrobial Susceptibility Testing; HR, heteroresistance; MIC, minimum inhibitory concentration; PAP, population analysis profiling.

For DAP-resistant mutants, mutations in *mprF* (3 out of 13 mutants) that encodes a lysyl-phosphatidylglycerol (LPG) synthase with flippase activity that contributes to an increase in the positive charge of the cell membrane [49], a mutation in *yvqF (vraT)* (1 out of 13 mutants), that acts as a transporter associated with the VraSR regulatory system, and a mutation in *prs* (1 out of 13 mutants) were found, all of which are known genes that confer DAP resistance in *S. aureus* clinical isolates when mutated [29,50–52] (Figs 2, 3 and S8). A deletion in the *rpsU* gene was frequently observed in a subset of mutants (4 out of 13), particularly in combination with other mutations. To our knowledge, there is no previous data showing that *rpsU* is involved in DAP resistance. Among the mutants with a single mutation, mutations were found in 2 tRNA ligases (*argS* and *thrS*) and glycerol metabolism (*glpD* gene), which similarly have not been previously associated with DAP resistance.

For GEN-resistant mutants, several genes were mutated (Figs 2, 3 and S9) that are involved in the menaquinone biosynthesis pathway (*menC*, *menE*, and *hepT*) or the chorismite biosynthesis pathway (*aroA*, *aroB*, and *aroK*), which is the precursor of the menaquinone pathway [53]. These mutations are all expected to reduce electron transfer and proton motive force which is a known mechanism of resistance towards aminoglycosides [54].

For OXA-resistant mutants, loss-of-function mutations in the *gdpP* gene coding for a phosphodiesterase enzyme involved in the degradation of *S. aureus* second messenger cyclic-di-AMP were present in the majority of mutants (16 out of 17 mutants) (Figs 2, 3 and S10). Mutations in *gdpP* is a well-known mechanism of resistance to β-lactam antibiotics in clinical methicillin-resistant *S. aureus* isolates [55].

Mutations occurring in different genes among mutants selected in presence of TEC were linked to (i) the peptidoglycan synthesis pathway (*vraR*, *vraS*, and *fmhC*) [56–58]; (ii) the cell membrane transport system, (*yvqF (vraT)* and *ywaC*) [57]; or (iii) the teichoic acid

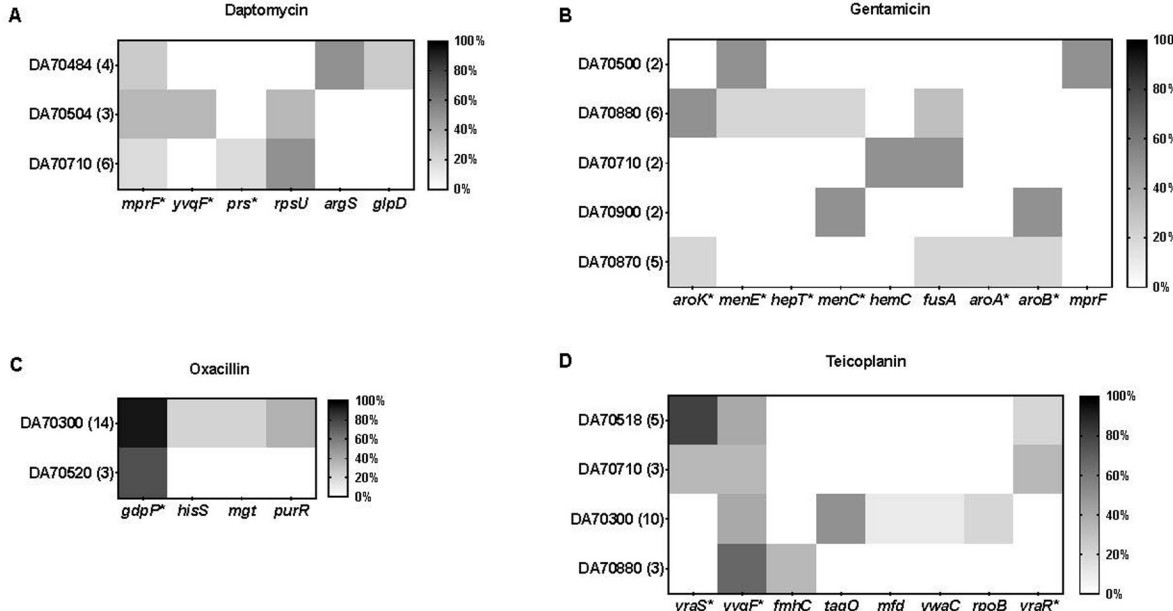

**Fig 2. Frequency of mutations in selected genes of heteroresistant strains.** Genes shown were selected based on (i) presence of mutations in multiple isolates or (ii) presence of mutations in genes known to confer resistance to the specific antibiotic when mutated in *S. aureus* (genes indicated by a star *). All mutations present in each isolate are presented in S8–S11 Figs. Strain numbers represent parental isolates and the total number of mutants for each parental isolate is shown in parenthesis. Light-to-dark gray scale represents the frequency of resistant mutants carrying a mutation in the same gene. (A) DAP (daptomycin), (B) GEN (gentamicin), (C) OXA (oxacillin), and (D) TEC (teicoplanin).

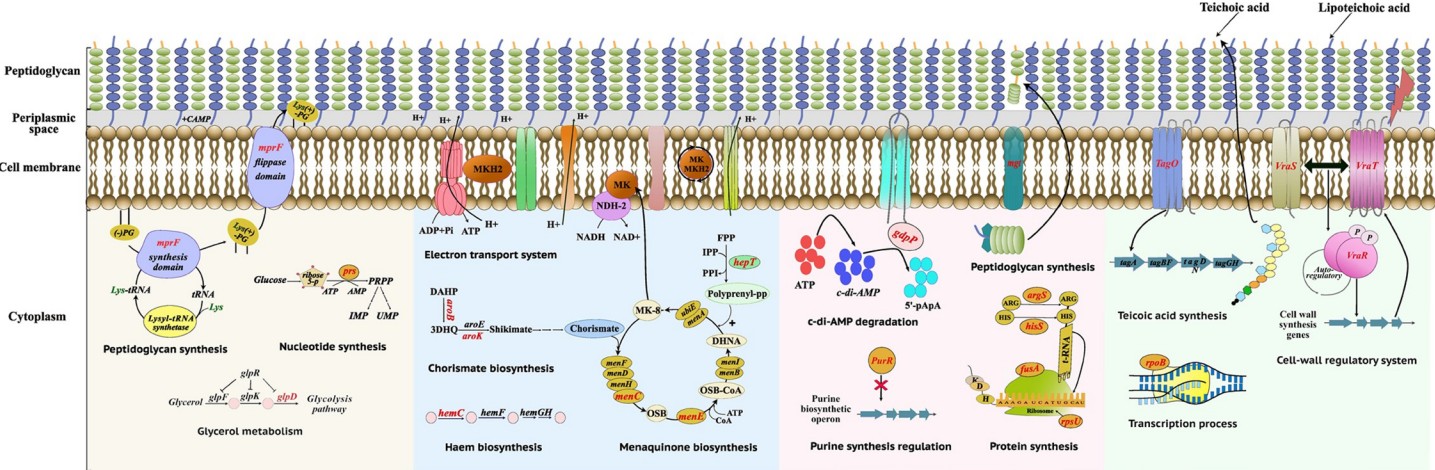

**Fig 3. Schematic representation of mutated pathways in resistant mutants.** The mutated genes identified within the resistant mutants were involved in a diverse range of pathways, including peptidoglycan synthesis (*mprF*, *mgt*, and *tagO*); protein synthesis (*argS*, *hisS*, *rpsU*, and *fusA*); glycerol metabolism (*glpD*); nucleotide synthesis (*prs*); chorismate and menaquinone biosynthesis (*aroB*, *arok*, *menC*, *menE*, and *hepT*); heme biosynthesis (*hemC*); regulation of c-di-AMP (*gdpP*); purine biosynthesis (*purR*); cell wall regulatory system (*vraS*, *yvqF* (*vraT*), *vraR*); and transcription process (*rpoB*). These genes are highlighted in red. Each antibiotic-associated gene group was designated a unique color-coded block except for mutated genes involved in protein synthesis. The color scheme included pale yellow for (DAP), blue for (GEN), pink for (OXA), and green for (TEC).

biosynthesis pathway (*tagO*) [59], which all are functionally linked to the TEC mechanism of action of inhibiting bacterial peptidoglycan synthesis (Figs 2 and 3 and S11) [60].

## Mutation rate analysis

To determine if a general increase in mutation rate (i.e., a mutator phenotype) could underlie the HR phenotype, a fluctuation assay was performed on all parental isolates to determine the mutation rate towards rifampicin resistance. Out of the 40 parental isolates, 39 had a similar mutation rate (average mutation rate $3.9 \times 10^{-9}$) while 1 HR isolate (DA70300) had a 10-fold increase in mutation rate compared to the average mutation rate of the other isolates. Mutants selected from this weak mutator isolate had 4 mutations on average as compared to the other HR isolates that had 1 to 2 mutations per mutant (Fig 4A and S1 Table). When comparing the mutation rate in non-HR isolates to isolates with HR to 1, 2, or 3 different antibiotics, there was no significant differences in the mutation rates (*p*-value 0.19) (Fig 4B and S1 Table). In conclusion, a general increase in mutation rate cannot explain the HR phenotype in our collection of isolates, except potentially for 1 single isolate.

## Discussion

In recent years, there has been a growing concern about the limited effectiveness of currently recommended antimicrobial agents in treating *S. aureus* infections, despite the observation that the isolates display MIC values within the susceptible range [61]. One potential contributing factor to this observation is bacterial strains with a HR phenotype. The lack of a consistent HR definition and diagnostic methods to detect HR, as well as limited knowledge about the genetic mechanisms underlying the HR phenotype has posed challenges for clinicians when it comes to identifying HR isolates and prescribing appropriate antibiotics. By using the standard definition and most common method for identifying HR, the PAP test, we observed a high prevalence of HR among the clinical *S. aureus* isolates studied. Importantly, these isolates showed susceptibility to the examined antibiotics using standard AST methods and their HR

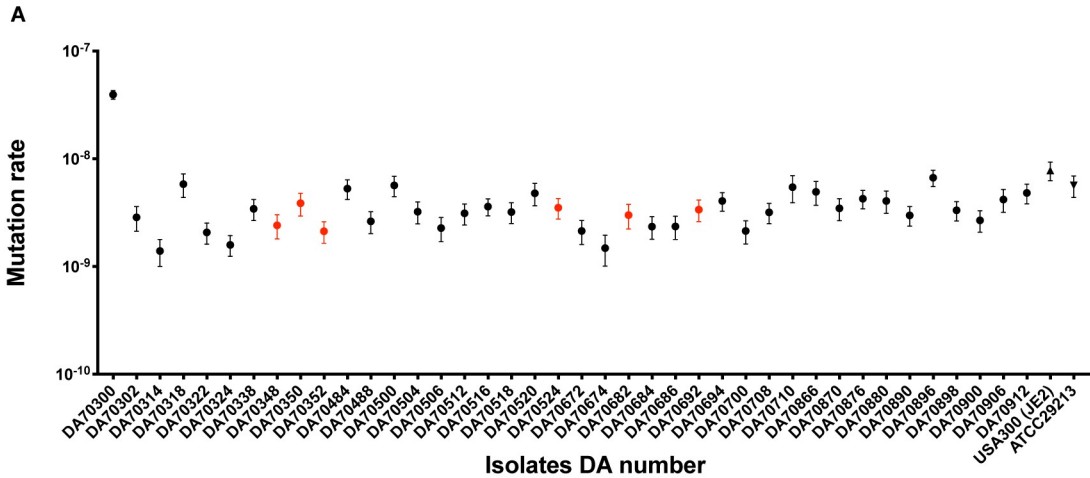

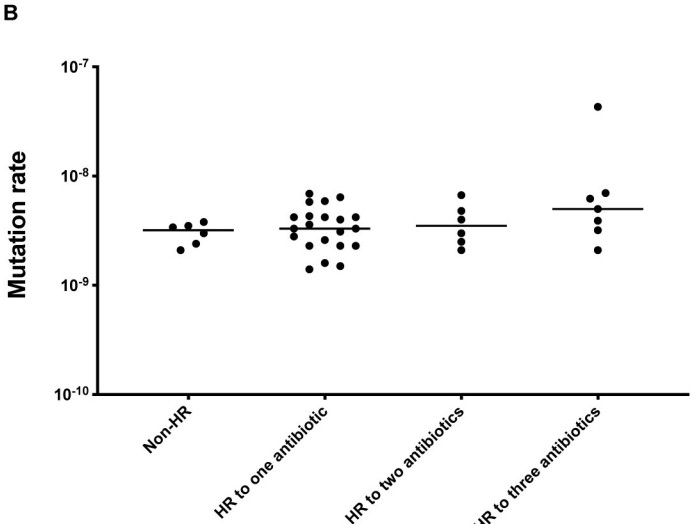

**Fig 4. Mutation rates for all parental isolates.** (**A**) Mutation rate analysis for all 40 parental *S. aureus* isolates. Mutation rates for rifampicin resistance were determined for each isolate based on 20 biological replicate cultures with 95% confidence intervals. *S. aureus* USA300 (JE2) and *S. aureus* ATCC 29213 were used as reference for the mutation rate analysis. Non-HR isolates are highlighted with red symbols while the HR isolates are colored black. (**B**) Mutation rates as a function of HR phenotype. For raw data, see S1 Table. HR, heteroresistance.

phenotype and potential associated risk of causing treatment failure would accordingly have been missed if we had relied solely on disk diffusion and E-tests.

The prevalence of HR varied for the different antibiotics with high frequencies for GEN, OXA, DAP, and TEC while none of the isolates showed HR toward LNZ or VAN. Why HR was not observed for LNZ and VAN is unclear but it could be because mutations conferring the resistance are rare and that the population size used for mutant selection ($10^8$ cells) is not sufficiently high to detect them. For GEN, OXA, DAP, and TEC (discussed below), many mutations appeared to be more frequent loss-of-function mutations (e.g., deletions, frameshifts, stop codons) and they are more likely to be found under the used selection conditions. Regarding stability, it can be noted that the observed point mutations are likely to be stable and reversion back to wild-type state would occur at very low rates. Whether compensatory mutations exist that can reduce the fitness costs was not tested.

It is noteworthy that according to EUCAST guidelines for 1/3 of the HR isolates the MIC of the resistant subpopulation surpassed the CBP for the antibiotic, indicating their potential clinical relevance (Table 2). However, this percentage will decrease if we apply the CLSI CBPs, as CLSI has higher CBP values for all antibiotics analyzed except DAP (Fig 1). Below we discuss the prevalence, the mutations found and the potential mechanisms involved in generating the resistant subpopulations.

## Daptomycin

In a study by Okado and colleagues, DAP HR was reported in clinical methicillin-resistant *S. aureus* (MRSA) at a frequency of 7.4% using PAP methods [51]. However, another study focusing on clinical MRSA isolates from 3 endocarditis patients found a much higher prevalence of DAP HR, nearly 30% [29]. Even though the same methods were employed in both studies as well as ours, a significant variation in the frequency of DAP HR phenotype was evident. This discrepancy could be attributed to differences in sample characteristics and geographic regions from which the samples were collected. Regarding our mechanistic study, functional enrichment analysis of mutated genes in the DAP-resistant mutants revealed that most of the mutations were associated with tRNA modification (*trmD*), tRNA aminoacylation (*argS*, *thrS*), and the ribosome (*rplV*, *rpsU*). In these mutants no mutations in the known DAP-resistance conferring *mprF* and *prs* genes were observed [29,62,63]. Thus, since mutations in *argS*, *thrS*, and *rplV* were the only mutations detected in several mutants they alone contribute to resistance (S8 Fig). A deletion in the *rpsU* gene was observed in some mutants, and although the role of this gene in resistance to DAP is unclear, some studies show that mutations in this gene affect the conversion of hVISA to VISA in clinical strains [64] and lead to reduced VAN susceptibility [65].

## Gentamicin

There was a particularly high HR occurrence observed for GEN. A high prevalence of GEN HR was also observed by Nicoloff and colleagues for gram-negative bacteria, especially for *Escherichia coli* and *Acinetobacter baumannii* [10]. Sequence analysis of GEN-resistant mutants showed that most mutations were located in genes associated with the menaquinone synthesis pathway, which plays a critical role in the cells electron transport system. Given that aminoglycoside transport is contingent on proton motive force, any mutations impacting the electron transport system (i.e., mutations affecting heme and menaquionone levels) are known to confer resistance to aminoglycosides [54,66]. Additionally, 1 resistant mutant had mutations in the *fusA* gene, resulting in a significant 4 to 5× increase in resistance to GEN (S6 and S9 Figs). Mutations in the *fusA* gene are known to provide resistance to fusidic acid in *S. aureus* [67,68]. This antibiotic targets the translation elongation factor EF-G in the ribosome complex, resulting in inhibition of protein synthesis similar to aminoglycosides [67]. Additionally, the involvement of the *fusA* gene in conferring resistance to GEN in *E. coli* is well established [69]. As some mutants carry mutations in the *fusA* gene, without any other known mutations associated with GEN resistance, it is likely that the *fusA* gene mutations confer the increased resistance to GEN. Mutations were also found in the *hemC* gene and since previous studies have described the association of mutations in the *hemH* gene with impaired electron transport and the emergence of aminoglycoside resistant small colony variants in *S. aureus* [70], it is plausible that the loss-of-function mutation in *hemC* confer resistance in a similar way in our mutants.

## Oxacillin

In the present study, OXA exhibited the second-highest prevalence of the HR phenotype. A similar significant prevalence of the OXA HR phenotype among 63 clinical isolates of *S. aureus* and 76 clinical isolates of coagulase-negative staphylococci was reported by Frebourg and colleagues at 53.6%. However, their study relied on another method (a large-inoculum OXA disk diffusion assay) to identify HR [71]. Our sequence analysis shows that aside from the well-known *gdpP* mutation, which occurs in 94% of all OXA-resistant mutants and is a recognized resistance mechanism [55], additional mutations were found in several genes across different isolates (S10 Fig). These genes include *mgt*, *purR*, and *hisS*, and their potential contribution in OXA resistance in *S. aureus* isolates is not known. Interestingly, the mutation in the *purR* gene has the potential to increase virulence of *S. aureus* by derepressing fibronectin binding proteins (FnBPs) and extracellular toxins, which are crucial for the development of a hypervirulent phenotype during the infectious process [72–74]. Furthermore, it is important to note that the *mgt* gene encodes a peptidoglycan polymerase, which plays a crucial role in catalyzing the elongation of glycan chains using lipid-linked disaccharide-pentapeptide as a substrate [75]. This aligns with the mechanism of action of OXA, which disrupts cell wall synthesis.

## Teicoplanin

HR for TEC in clinical isolates of *S. aureus* has been previously documented. In the study performed by Nakipoglu and colleagues, the occurrence of HR for TEC among *S. aureus* strains was notably low, at 1.2%, whereas it was 13% among methicillin-resistant coagulase-negative *S. aureus* [76]. Additionally, when considering other bacterial species, particularly *Enterecoccus* spp., the prevalence of TEC HR is significantly higher than in *S. aureus* isolates. A study by Park and colleagues reported a close to 100% prevalence of TEC HR in an outbreak of VanB phenotype-*vanA* genotype *Enterococcus* isolates obtained from 6 patients at a hospital in Korea [77]. This high prevalence may be attributed to the different method (E-test) used and the limited number of isolates they examined. In our study, genome sequencing of the resistant subpopulation for TEC HR isolates showed that the majority of resistant mutants (15 out of 21, 72%) harbored mutations in one of the following genes: *vraR*, *vraS*, and *yvqF (vraT)*, which are known to confer resistance to TEC [57,58]. The remaining isolates (5 out of 21, 24%) had mutations in the *tagO* gene, which plays a role in teichoic acid synthesis as part of the cell membrane [59]. One resistant mutant had a mutation in the *tcaA* gene, and previous work has linked mutations in *tcaA* to TEC resistance [78]. This isolate also had a mutation in the *vraR* gene. Most of the TEC-resistant mutants also carried additional mutations in other genes (S11 Fig), but at present it is unclear if these mutations also contribute to the resistance phenotype.

## Linezolid and vancomycin

In our set of clinical isolates, no HR isolates were found for LNZ or VAN. As a result, no resistant mutant selection or mechanistic analysis was conducted for these 2 antibiotics. The absence of HR in these 2 antibiotics may be attributed to differences in the HR identification procedure, the definitions for HR used in other studies, or to the specific isolates analyzed. For example, in a study conducted by Wootton and colleagues, the prevalence of hVISA isolates varied from no HR to 7% using gradient plates and standard PAP methods [34]. Similarly, in another study by Sancak and colleagues, the occurrence of hVISA was reported as 13.7% using another method with different criteria than the ones we employed. In 2 other studies, hVISA prevalence was 6% to 8% using the macro E-test (MET) method [26,79]. In addition, based on the meta-analysis, the global prevalence of HR for VAN appears to be low, less than 6% [80], which aligns with our findings for this antibiotic. Additionally, due to the lack of correlation

between the prescreening approach and the PAP test for these antibiotics, there is a possibility that we may have missed true cases of HR. Regarding LNZ, it is important to note that the most common mechanism of resistance involves nucleotide substitutions in the central domain of the V region of 23S rRNA and the frequency of this mutational resistance is rare and requires prolonged exposure to the antibiotic [81]. Consequently, our approach, with incubation for 72 h, may not be able to detect the HR phenotype. Moreover, achieving a high level of resistance to LNZ involves mutations in all copies of the 23S rRNA gene [81]. Considering these factors, along with the low worldwide occurrence of LNZ resistance [82], the likelihood of detecting HR for this antibiotic among the 40 isolates examined here is likely low.

## Differences between *S. aureus* and gram-negative bacteria in resistance mechanisms

In our study, the main contributing mechanism of resistance in *S. aureus* involved chromosomal point mutations in different genes. This finding contrasts with previous studies conducted in 4 gram-negative bacterial species, which showed that tandem amplification of resistance genes was the primary cause of the HR phenotype, where an increased dosage of the resistance-conferring enzyme resulted in reduced susceptibility [6,83]. However, chromosomal mutations were also observed for some gram-negative species and in particular for aminoglycosides [10]. An interesting question is why gene amplification of resistance genes was not observed in our *S. aureus* isolates? As shown by an analysis of the resistance gene content in our strains, the absence of amplification as the cause of HR in most of our isolates might be attributed to the lack of bona fide resistance genes that can confer resistance to the antibiotic used for mutant selection (S4 Table). Thus, only a few isolates, DA70300, DA70520, and DA70870, carried a known resistance gene that could potentially confer resistance to the antibiotic on which they were selected if increased in copy number (S4 Table).

The role of the *APH(3')-IIIa* gene, as found in the DA70870 isolate, in providing resistance to aminoglycoside antibiotics such as kanamycin and streptomycin in *S. aureus* and *Enterococci* is well established [84,85]. Furthermore, among aminoglycoside-modifying enzymes, this particular type is less prevalent in *S. aureus* isolates compared to other genes such as *ac(6')-Ie-aph(2″)* when it comes to conferring resistance to aminoglycosides in MRSA isolates [86]. There is currently no documented evidence showing that the presence of *APH(3')-IIIa* may confer resistance to GEN.

The genomic sequences of isolates DA70300 and DA70520 revealed the presence of the *mgrA* and *blaZ* genes, which are known to be involved in regulating autolysis and contribute to resistance to cell wall-active antibiotics such as OXA. It has been observed that altering *mgrA* gene expression in response to sub-MIC exposure of β-lactam antibiotics can result in OXA resistance in HR MRSA isolates [33]. However, the effectiveness of *mgrA* is more pronounced at low concentrations of OXA (sub-MIC) [87], which were not used in our mutant selection (which was done at 8 times the MIC). The *S. aureus* β-lactamase (*blaZ*) genes were found in both DA70300 and DA70520 isolates. The BlaZ proteins are serologically classified into type A to D [88] and studies show that the β-lactamase activity of different types of BlaZ protein can vary depending on serological type. One study showed that class A β-lactamase (penicillinase) exhibits lower hydrolytic activity against OXA, cephems, and carbapenems compared to types B, C, and D [89], whereas another study demonstrated that type A Staphylococcal β-lactamase functions as an extended-spectrum enzyme, contributing to borderline resistance to OXA in *S. aureus* [90]. Yet another study suggested that type D B-lactamase displays notably slower hydrolysis of penicillin than the other types [88]. Additionally, the most prevalent and understood type of BlaZ protein in *S. aureus* are the plasmid-located types A, C,

and D. Since our 2 strains carry the type B *blaZ* gene located on the chromosome, which is uncommon and not as well studied, it is difficult to assess whether amplification of this gene could confer resistance to OXA.

Gene amplification is initiated by formation of a duplication that is generated by recombination between directly repeated DNA sequences (e.g., IS elements, transposase genes, rRNA operons) that happen to flank the resistance gene. This duplication can then be further amplified or lost depending on whether selection is present for the resistance gene. Thus, for gene amplification to occur and provide HR it is required that both a resistance gene is present and that it is flanked by repeat sequences that provide substrates for recombination. The flanking repeats are typically located less than 200 kbp from the resistance gene [10,20]. Thus, we searched for potential repeat sequences around the *APH(3')-IIIa*, *mgrA*, and *blaZ* genes in a region stretching 200 kbp on each side of the respective gene. This analysis showed the absence of direct repeats surrounding the relevant resistance genes which would reduce the likelihood of gene amplification [2,10]. In summary, for *S. aureus* HR isolates the dominance of single-nucleotide polymorphisms and small insertions/deletions in generating resistance [50,51,91–93], and the absence of gene amplification, can be explained by a combination of a generally low content of resistance genes and absence of surrounding repeat sequences as compared to gram-negative species.

Why certain isolates show an HR phenotype (i.e., a frequency of resistant mutants that reaches the $\geq 10^{-7}$ threshold and a $\geq$8-fold increase in resistance) is a key question. One possibility is that HR isolates have a generally higher mutation rate (i.e., a mutator phenotype), which was observed in another study involving *E. coli* isolates with regard to fosfomycin resistance [94]. However, in our study the rifampicin resistance mutation rate (used as a proxy for measuring the general mutation rate) was largely similar when comparing non-HR and HR isolates. Thus, in our set of isolates, there was no link between the mutation rate and having an HR phenotype (Fig 4A and 4B). Another potential explanation is that HR strains have a higher MIC to the specific antibiotic for which they show the HR phenotype, and as a result it might be easier to achieve a mutational increase of MIC above the $\geq$8-fold threshold. This explanation is unlikely though since there was no correlation between the MIC value and whether the strains were classified as HR or non-HR (S5 Fig). In fact, the HR isolates in our study exhibited significantly lower MIC values compared to the non-HR isolates for DAP and TEC (S5 Fig) which is in contrast to previous studies that suggested a relationship between higher MIC values and the presence of the hVISA phenotype [26]. Furthermore, we observed that some isolates with the highest MIC values did not demonstrate HR towards any of the antibiotics examined (such as DA70350) or towards specific antibiotics (such as DA70900 for TEC) (Tables 1 and S3). One final potential explanation (that was not examined here) is that strain-specific epistatic effects influence the rate of mutant emergence. For example, if the fitness costs and/or resistance level conferred by specific resistance mutations vary depending on genetic background, in strains where the mutations confer a high cost and a low resistance the mutation rate would appear low and they would be classified as non-HR strains. In contrast, for strains where costs are low and/or resistance level reached are high, the mutation rate would appear to be higher and they would be classified as HR strains. Previous work show that such epigenetic effects are common for certain types of resistance mutations [95] but further genetic analysis is required to test this hypothesis.

## Conclusions

This study shows that HR to several clinically relevant antibiotics is highly prevalent in *S. aureus* and that regular chromosomal mutations in various core genes is the main mechanism

by which HR is conferred. The reason for the lack of gene amplification mechanisms, which is the main mechanism observed in several gram-negatives, appears to be a lack of resistance genes and repeat sequences that allow their amplification in *S. aureus*. The high prevalence of HR among *S. aureus* isolates highlights the urgent need for further investigation into HR detection and prevalence, the HR-conferring genetic mechanisms and the potential impact of these resistant subpopulations on treatment. The latter is particularly important since many of the HR isolates show a resistant subpopulation with an MIC above the CBP that is expected to better resist drug concentrations achieved in a treated patient, and since several isolates were also HR to several antibiotics.

## Supporting information

**S1 Fig. Frequency and distribution of MIC values of 6 different antibiotics for 40 clinical *S. aureus* isolates.** MICs were measured with a single E-test. The dotted vertical lines represent the threshold for clinical resistance according to EUCAST. (A) MICs of DAP (daptomycin), (B) MICs of GEN (gentamicin), (C) MICs of LNZ (linezolid), (D) MICs of OXA (oxacillin), (E) MICs of TEC (teicoplanin), and (F) MICs of VAN (vancomycin).
(PDF)

**S2 Fig. Schematic representation of the prescreening procedure.**
(PDF)

**S3 Fig. Frequency of susceptible *S. aureus* isolates growing at antibiotic concentrations (4×, 8×, and 16×) above MIC determined in the prescreen for each antibiotic.** Isolates showing growth of subpopulations at 8- or 16-fold the MIC value were selected for PAP tests to confirm an HR phenotype. The total number of susceptible isolates used for prescreening of each antibiotic was: DAP (daptomycin): 39, GEN (gentamicin): 39, LNZ (linezolid): 40, OXA (oxacillin): 37, TEC (teicoplanin): 39, and VAN (vancomycin): 40.
(PDF)

**S4 Fig. Population analysis profile (PAP) of all isolates identified as probable HR in the prescreen.** Each curve is based on the average of 3 independent experiments. The red line on the graph marks the non-HR isolates. The horizontal dotted lines indicate the subpopulation frequency cutoff ($1 \times 10^{-7}$) used to identify HR isolates. For raw data, see S2 Table.
(PDF)

**S5 Fig. Distribution of HR and non-HR isolates of *S. aureus* as a function of MIC value.** Symbols ** and **** indicate high significance in the Mann–Whitney test, with the *p*-values of <0.0001 and 0.0073, respectively, while "ns" shows nonsignificant differences. (A) DAP (daptomycin), (B) GEN (gentamicin), (C) OXA (oxacillin), and (D) TEC (teicoplanin).
(PDF)

**S6 Fig. Relative growth rate and resistance profile for parental strains and all resistant mutants belonging to each parental strain.** DA numbers for each parental strain and mutant are on the x-axis. The relative growth rates (normalized to the growth rate of the parental strain) and MIC values are represented with bars and dots, respectively. Parental isolates (dark bars and white dots) are always followed by the mutants (gray bars and black dots) isolated from that specific parental isolate. Relative growth rates are based on 5 biological replicates and the error bars indicate standard deviation. MICs are based on a single E-test. (A) DAP (daptomycin), (B) GEN (gentamicin), (C) OXA (oxacillin), and (D) TEC (teicoplanin).
(PDF)

**S7 Fig. Correlations between relative growth rate and resistance level.** Mutants and parental isolates are indicated with black and red dots, respectively. (A) DAP (daptomycin), (B) GEN (gentamicin), (C) OXA (oxacillin), and (D) TEC (teicoplanin).
(PDF)

**S8 Fig. Mutations in DAP (daptomycin)-resistant mutants.** Mutants (DA number below) belong to parental isolates indicated on top of the graph. Fs indicates frame shift, * stop-codon and Δ deletion.
(PDF)

**S9 Fig. Mutations in GEN (gentamicin)-resistant mutants.** Mutants (DA number below) belong to parental isolates indicated on top of the graph. Fs indicates frame shift and * stop-codon.
(PDF)

**S10 Fig. Mutations in OXA (oxacillin)-resistant mutants.** Mutants (DA number below) belong to parental isolates indicated on top of the graph. Fs indicates frame shift, * stop-codon and Ins insertion of nucleotide.
(PDF)

**S11 Fig. Mutations in TEC (teicoplanin)-resistant mutants.** Mutants (DA number below) belong to parental isolates indicated on top of the graph. Fs indicates frame shift, * stop-codon and Dup, duplication.
(PDF)

**S1 Table. Raw data for Figs 4 and S1, S3, S6 and S7 (in separate excel file).**
(XLSX)

**S2 Table. CFU data for PAP analysis of DAP, GEN, LNZ, OXA, TEC, and VAN (in separate excel file).**
(XLSX)

**S3 Table. Frequency of HR for each of the 40 clinical *S. aureus* isolates to 6 different antibiotics.**
(PDF)

**S4 Table. Antibiotic resistance genes in heteroresistant isolates used for selection and whole-genome analysis of mutants.**
(PDF)

## Acknowledgments

We would like to thank Örjan Samuelson (University Hospital North Norway, Tromsö, Norway), Niels Frimodt-Möller (Rigshospitalet, Copenhagen, Denmark), Volkan Özenci (Karolinska University Hospital, Solna, Sweden), and Fernando Baquero (Hospital Ramon y Cajal, Madrid, Spain) for providing us with the *S. aureus* clinical isolates.

## Author Contributions

**Conceptualization:** Dan I. Andersson.

**Formal analysis:** Sheida Heidarian, Karin Hjort.

**Investigation:** Sheida Heidarian, Andrei Guliaev.

**Methodology:** Sheida Heidarian, Andrei Guliaev, Hervé Nicoloff, Karin Hjort, Dan I. Andersson.

**Resources:** Dan I. Andersson.

**Supervision:** Hervé Nicoloff, Karin Hjort, Dan I. Andersson.

**Writing – original draft:** Sheida Heidarian.

**Writing – review & editing:** Sheida Heidarian, Andrei Guliaev, Hervé Nicoloff, Karin Hjort, Dan I. Andersson.

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
