## [Editor Report · Decision Letter 0]

19 Sep 2023

Dear Dr. Andersson, 

Thank you for submitting your manuscript entitled "High prevalence of heteroresistance in Staphylococcus aureus is caused by a multitude of mutations in core genes" for consideration as a Research Article by PLOS Biology.

Your manuscript has now been evaluated by the PLOS Biology editorial staff, as well as by an academic editor with relevant expertise, and I am writing to let you know that we would like to send your submission out for external peer review as a Short Report. Please select this in the submission system where corresponds when re-submitting. 

Once your full submission is complete, your paper will undergo a series of checks in preparation for peer review. After your manuscript has passed the checks it will be sent out for review. To provide the metadata for your submission, please Login to Editorial Manager (https://www.editorialmanager.com/pbiology) within two working days, i.e. by Sep 21 2023 11:59PM.

Kind regards,

Paula

---

Senior Editor

PLOS Biology

---

## [Decision Letter · Decision Letter 1]

27 Oct 2023

Dear Dr. Andersson,

Thank you for your patience while your manuscript "High prevalence of heteroresistance in Staphylococcus aureus is caused by a multitude of mutations in core genes" went through peer-review at PLOS Biology. Your manuscript has now been evaluated by the PLOS Biology editors, an Academic Editor with relevant expertise, and by several independent reviewers.

In light of the reviews, which you will find at the end of this email, we are pleased to offer you the opportunity to address the comments from the reviewers in a revision that we anticipate should not take you very long. We will then assess your revised manuscript and your response to the reviewers' comments with our Academic Editor aiming to avoid further rounds of peer-review, although might need to consult with the reviewers, depending on the nature of the revisions.

In particular, we think it is important that you clarify the definition of heteroresistance, the stability of the resistant subpopulations, and the comment of reviewer #1 about the different dilutions used before the PAP test. Please address all the reviewers' comments.

**IMPORTANT - SUBMITTING YOUR REVISION**

*Resubmission Checklist*

*Published Peer Review*

***PLOS Data Policy***

**Please note that as a condition of publication PLOS' data policy (http://journals.plos.org/plosbiology/s/data-availability) requires that you make available all data used to draw the conclusions arrived at in your manuscript. If you have not already done so, you must include any data used in your manuscript either in appropriate repositories, within the body of the manuscript, or as supporting information (N.B. this includes any numerical values that were used to generate graphs, histograms etc.). For an example see here: http://www.plosbiology.org/article/info:doi%2F10.1371%2Fjournal.pbio.1001908#s5**

*Blot and Gel Data Policy*

Sincerely,

Paula

---

Senior Editor

PLOS Biology

REVIEWS:

Reviewer #1: Protein evolution.

Reviewer #2: Antibiotic resistance.

Reviewer #1: This manuscript explores the mechanisms of heteroresistance, a relatively underexplored phenomenon in which a sensitive population consists of a small subpopulation of unstable resistant variants, across forty clinical Staphylococcus strains exposed to a variety of antibiotics. The authors reveal varying degrees of heteroresistance against gentamicin, oxacillin, daptomycin, and teicoplanin, with heteroresistance against gentamicin being particularly prevalent. The authors then go on to identify the mechanisms responsible for this heteroresistance by whole-genome sequencing selected resistant mutants from heteroresistant subpopulations, and found that mutations in different core chromosomal genes were responsible for generating these mutants. Finally, they investigate whether mutator phenotypes could be responsible for increased heteroresistance levels in Staphylococcus using fluctuation tests, but found no supporting evidence for this.

The topic of this study would be of interest to clinical microbiologists, particularly because information on heteroresistance in gram-positive bacteria is relatively limited. The manuscript is well-written, and the figures are appropriately employed. The methods and statistical analyses are clear and generally appropriate.

Comments

My main comment pertains to the definition of heteroresistance. I have consistently understood that a defining feature of heteroresistant variants is their high instability, with a tendency to revert back to susceptibility in the absence of selective pressure. This characteristic is also highlighted in the abstract and introduction of the manuscript. Therefore, I believe it is essential to demonstrate this instability in the putative heteroresistant variants. Although the authors have explored the fitness costs associated with these mutants and have found that some may exhibit significant fitness costs (which might facilitate rapid reversion), while others show little to no fitness cost, this alone doesn't establish the unstable nature of these variants.

If the authors argue that heteroresistant variants can also be stable, it becomes imperative to differentiate heteroresistant variants from regular resistant variants that originate from spontaneous mutations, either during the PAP analysis or more likely during the pre-culture. From my perspective, as an evolutionary biologist with a focus on resistance evolution, it seems that the authors have not identified true heteroresistant mutants but rather typical resistant mutants. Consequently, the variations in heteroresistance levels merely result from differences in mutation supply and/or target size for resistance to the specific antibiotic in question. In summary, the need to clearly distinguish between heteroresistance and regular resistance, particularly by demonstrating the instability of putative heteroresistant variants, is essential for a more accurate interpretation of the findings. If there is no difference, then I don't really understand the need for invoking heteroresistance.

In the methods section discussing the PAP test, the authors mention that a 10-2 or 10-3 dilution was transferred to 1ml MHB (to generate a pre-culture from which samples would be taken for the PAP test). I would like to understand the reason behind this difference in dilution. It's worth noting that the additional generation in the 10-3 samples could potentially result in a higher number of variants, which might contribute to increased heteroresistance levels. Therefore, clarification on the rationale for this difference in dilution is warranted.

Raw data from the PAP tests or whole-genome sequencing is not available.

Minor comments/ suggestions:

Fig S4: show the individual replicate data points or mean with error.

Fig S1: I would recommend transforming this scatterplot into a barplot for improved clarity. Additionally, adding the names of the antibiotics either within or above each corresponding panel would enhance the reader's experience, reducing the need to repeatedly refer to the legend (also for Fig 1, 2 and Fig S7 this would be helpful).

Reviewer #2: In this study, Heidarian and colleagues assess the potential for clinical isolates of Staphylococcus aureus to generate - in non-selective culture - a suite of mutants which are resistant to antibiotics. Both the frequency of these mutants in the population and the levels of resistance expressed by these mutants are high enough that many experimental populations can be identified as having "heteroresistance" (using standardized methods and categorizations). Resistant mutants were isolated, sequenced and the mutations conferring resistance were identified (but not directly tested for causality). Mutations underpinning the sub-population of resistant types were found across core genes. In gram-negative bacteria, such mutations might be found in genes which primarily degrade, alter or export the target antibiotic, or via amplification of such genes. The authors conclude that the mutational profile of mutations likely-causal of heteroresistance in these populations of S. aureus is probably due to 1) the lack of resistance genes in these isolates and 2) the lack of repetitive regions which would allow amplification of such genes.

I'm not very familiar with the HR literature, so I can't really comment as to the novelty or significance of the findings. The manuscript describes experiments with a suitable level of rigor, and the conclusions are reasonably supported with evidence. It is generally best to reconstruct mutations to prove functional causality, and the claims about the distinction of this gram positive and negative strain are made with reference to the literature, and are not directly tested. However, I'm not convinced such thoroughness is required for a brief report and I won't demand further experiments. Similarly, I'd like to see some points made more explicit or clarified (see below), but I won't demand extensive additions.

One thing that does strongly need attention is a lack of data availability (see point 1 below). PLOS does have a clear data policy "All data and related metadata underlying reported findings should be deposited in appropriate public data repositories, unless already provided as part of a submitted article". There is no raw genomic data provided in the supplementary files, which should really be provided. Reproducibility of most of the findings is contingent on having exactly these strains and it's a bit of a push to expect they are shared considering they are pathogenic. Regardless, the authors should at least provide the raw sequencing data so claims about the lack of resistance genes/repetitive regions in these strains can be independently assessed.

Subsequent points (2-7) describe areas that require attention to avoid confusion from a reader's perspective.

1. It's claimed in the author statements that all raw data are available in the supplementary files, however, considerable amounts of raw genomic data are obviously not present. I assume that the raw dataset (raw genomic sequencing files) will be uploaded to a repository/genbank upon publication of this MS. This is important, especially so readers could assess the claims about the lack of repetitive regions in S. aureus. Very helpful files to include in this dataset would be the reference sequence made of the parental isolates.

2. There are several references to "stability" in the abstract and introduction (paragraphs 1 and 3). This theme of stability is not explicitly returned to in the discussion. I assume the HR-causing mutations described for these population of S. aureus are stable (and require secondary mutation to allow compensation of their deleterious effects in the absence of selection). Please explicitly discuss the "stability" of the mutations seen here (i.e. the likelihood that the exact mutation reverts to WT) and what this means for the subsequent evolution of S. aureus (i.e. are compensatory mutations expected). Alternatively, remove the stability concept from the introduction and abstract.

3. I was confused on the first reading with the various references to "Isolates". The results start with a description of "isolates", and I misinterpreted this to mean an analysis of HR isolates (not parental clonal isolates). Please more strongly distinguish parental isolates and HR isolates.

4. Similarly, it is helpful for the reader to realize (early in the results) how the HR types were generated without reference to the methods. For those unfamiliar with HR, it was not obvious in what conditions these HR types evolved. The diversity of HR types is generated from a limited population size with relatively few generations (<20 generations) in the absence of selection. I suggest it is made clearer that the HR types evolved under such conditions. This would allow a better understanding of the conduct of the experiments, and makes the high diversity of HR types even more surprising.

5. There is repeated reporting of the "prevalence frequencies" (in the abstract, introduction, results section "Prevalence of HR varies depending on antibiotic", the Y-axis of Fig S3 etc). The prevalence of the isolates is potentially misleading. Reported is the prevalence of cultures within which HR types are identified. It could be interpreted that the prevalence is the proportion of HR types within populations (as shown in Figure 1, Y-xis.). Please be more explicit that the "prevalence frequency" means the "frequency of cultures with HR".

6. While there is a very detailed description of the predicted function of each mutant locus and how this may provide resistance to each AB, lacking is a more general discussion of the classes of mutations selected by the antibiotics. It appears that loss-of-function mutations are generally more prevalent, and as partially suggested in the discussion, it seems LNZ and VAN cannot be caused by such mutations. A more general interpretation of the data is a that HR is made possible if large numbers of resistant mutants are generated before plating on selective antibiotics (ie, the large target size of these target genes aids the production of HR types). I'd suggest introducing this framing in your description of these mutations as it might provide a broader appeal to those interested in genetics.

7. Section "Differences between S. aureus and Gram-negative bacteria in resistance mechanisms". The authors hypothesize that the presence of resistance of resistance genes alters the mechanism by which HR occurs in gram-negative bacteria. However, there's no reason to believe mutations to core genes (as described here) is not occurring in gram-negative species, but they are rarely selected because "better" solutions (amplification of resistance genes) are found. It would be an idea (which the authors do not need to implement) to directly state this, which allows the testable predication that HR-mutants are to be expected in gram negative bacteria which lack resistance genes.

---

## [Editor Report · Decision Letter 2]

17 Nov 2023

Dear Dr. Andersson,

Thank you for your patience while we considered your revised manuscript "High prevalence of heteroresistance in Staphylococcus aureus is caused by a multitude of mutations in core genes" for publication as a Short Reports at PLOS Biology. This revised version of your manuscript has been evaluated by the PLOS Biology editors and the Academic Editor.

Based on our Academic Editor's assessment of your revision, we are likely to accept this manuscript for publication, provided you satisfactorily address the remaining points raised by the Academic Editor regarding the definition of Heteroresistance. You will find the comments at the end of this letter.

Please also make sure to address the following data and other policy-related requests.

1. DATA POLICY:

A) Supplementary files (e.g., excel). Please ensure that all data files are uploaded as 'Supporting Information' and are invariably referred to (in the manuscript, figure legends, and the Description field when uploading your files) using the following format verbatim: S1 Data, S2 Data, etc. Multiple panels of a single or even several figures can be included as multiple sheets in one excel file that is saved using exactly the following convention: S1_Data.xlsx (using an underscore).

B) Deposition in a publicly available repository. Please also provide the accession code or a reviewer link so that we may view your data before publication.

Regardless of the method selected, please ensure that you provide the individual numerical values that underlie the summary data displayed in the following figure panels as they are essential for readers to assess your analysis and to reproduce it: Figures 1ABCDEF, 2ABCD, 4AB, and Supplementary Figures S1, S3, S4, S5ABCD, S6ABCD, S7ABCD, S8, S9, S10, S11.

**Please also ensure that figure legends in your manuscript include information on where the underlying data can be found, and ensure your supplemental data file/s has a legend.**

2. CODE POLICY

Per journal policy, as the code that you have generated is important to support the conclusions of your manuscript, we require that you make it available without restrictions upon publication. Please ensure that the code is sufficiently well documented and reusable, and that your Data Statement in the Editorial Manager submission system accurately describes where your code can be found.

We expect to receive your revised manuscript within two weeks.

*Published Peer Review History*

*Press*

Sincerely,

Paula

---

Senior Editor,

pjaureguionieva@plos.org,

PLOS Biology

EDITED COMMENTS FROM THE ACADEMIC EDITOR:

Authors need to clarify the broad definition of HR (including instable gene amplifications and stable SNPs and other mutations) upfront in the abstract and first paragraph of the introduction. There, they raise the kind of confusion that bothered also rev#1, by saying HR is defined by the presence of "small subpopulations of resistant and often unstable cells". In their response to this comment, the authors explained very clearly what HR entails. I therefore suggest the authors to use their extensive and clear response to this comment of rev#1 to clarify the definition of HR in abstract and introduction. Basically, they should say HR is defined by the presence of sufficiently large (i.e. >10^-7) subpopulations of sufficiently resistant (i.e. >8-fold the MIC of wildtype) bacteria, whose genotypes have found to be often unstable and involve gene amplifications in Gram negative species.

---

## [Editor Report · Decision Letter 3]

30 Nov 2023

Dear Dr Andersson,

Thank you for the submission of your revised Short Reports "High prevalence of heteroresistance in Staphylococcus aureus is caused by a multitude of mutations in core genes" for publication in PLOS Biology. On behalf of my colleagues and the Academic Editor, Arjan de Visser, I am pleased to say that we can in principle accept your manuscript for publication, provided you address any remaining formatting and reporting issues. These will be detailed in an email you should receive within 2-3 business days from our colleagues in the journal operations team; no action is required from you until then. Please note that we will not be able to formally accept your manuscript and schedule it for publication until you have completed any requested changes.

PRESS

Sincerely, 

Paula 

---

Senior Editor

PLOS Biology
